# Human Decision-Making in Crowds in a Virtual Flood Scenario

Booloot Arshaghi <sup>1,\*</sup>, Glyn Lawson <sup>2</sup>, Riccardo Briganti <sup>3,</sup> Peer-Olaf Siebers <sup>4</sup>

- <sup>1</sup> University of Nottingham, Faculty of Engineering, Human Factors Research Group, Nottingham, UK, NG7 2RD.
- <sup>2</sup> Nottingham Trent University, School of Art and Design, Waverley Building, NG7 4HF.
- <sup>3</sup> University of Nottingham, Faculty of Engineering, Environmental Fluid Mechanics and Geo-processes Research Group, Coates Building, University Park, Nottingham, UK, NG7 2RD
- <sup>4</sup> University of Nottingham, School of Computer Science, Jubilee Campus, Wollaton Road, Nottingham, UK, NG8 1BB.

Correspondence to: Booloot Arshaghi (egxbel@nottingham.ac.uk)

Abstract. Flood evacuation outcomes are critically shaped by human behaviour, yet empirical data on individual decision-making remain scarce due to the dangers and logistical challenges of collecting data during real disasters. To address this gap, this study used Virtual Reality (VR) to examine how social cues, specifically crowd behaviour, interact with factors such as crowd size, clarity of the safe destination, and floodwater level to influence evacuation choices and delays. Four within-subjects VR experiments were conducted with 84 participants, systematically testing these variables in an immersive flood scenario. Results showed that crowd behaviour strongly determined both route choice and evacuation latency, often outweighing other factors. Participants tended to follow crowds into floodwater, demonstrating herding behaviour. However, this influence weakened when water levels were very high, indicating a threshold where physical danger overrides social cues. Larger crowds and unclear destination information further increased reliance on social information and pre-movement times. These findings highlight the powerful role of social dynamics in emergency decision-making and underscore the need to integrate realistic human behaviour, particularly social influence, into flood risk models, public warnings, and evacuation planning to improve community resilience and safety.

**Keywords:** Flood; Human Behaviour; Evacuation; Virtual Reality.

# 1 Introduction

Floods are among the most devastating natural disasters, causing widespread fatalities and displacement globally (Petrucci 2022). Intensified rainfall from climate change (Arias, Bellouin et al. 2021) and increasing population density in flood-prone areas (Ferdous, Di Baldassarre et al. 2020) have heightened long-term risks to communities. As global displacement rises (Menne, Murray et al. 2013) prioritising climate adaptation and risk mitigation strategies, such as evidence-based evacuation planning, has become essential (Aerts, Botzen et al. 2018, Alonso Vicario, Mazzoleni et al. 2020). Moreover, individuals often lack awareness or experience with extreme emergencies like flooding, leading to inadequate or inappropriate responses

55

(Mol, Botzen et al.). As a result, evacuation decisions may undermine risk mitigation efforts. For example, people may voluntarily enter floodwater for reasons such as continuing daily activities, fulfilling work obligations, retrieving belongings, or even for recreational purposes (Becker, Taylor et al. 2015, Petrucci 2022).

Traditional flood risk management has focused on hazard, exposure, and vulnerability assessments, including flood 35 likelihood and structural interventions (Alonso Vicario, Mazzoleni et al. 2020), supported by well-established hydraulic models (Nkwunonwo, Whitworth et al. 2020). However, these approaches often overlook human responses during emergencies, which critically influence outcomes such as fatalities (Simonovic and Ahmad 2005, Hamilton, Demant et al. 2020, Du, Wu et al. 2023). Recent research increasingly integrates human behaviour into flood modelling, offering a more comprehensive understanding of flood risk (Aerts, Botzen et al. 2018, Zhang, Liu et al. 2024, Shi, Li et al. 2025). While the 40 simulation of human behaviour in emergencies has a long history in fire evacuation research, which captures complex behavioural patterns (e.g., (Ronchi and Nilsson 2013, Ronchi 2021)), flood-specific behavioural modelling remains less developed. The flood domain lacks a clearly defined framework integrating human behaviour (e.g., the influence of social cues or socio-demographics), due to limited empirical data and insufficient understanding of how individuals interact with flood hazards, environments, and others (Aerts 2020, Alonso Vicario, Mazzoleni et al. 2020, Zhuo and Han 2020, Irsyad and Hitoshi 2022, Petrucci 2022). Existing models often oversimplify behaviour, ignoring key social and psychological dimensions (Shirvani and Kesserwani 2021). One such factor is the influence of crowd behaviour, which can significantly shape an individual's perception of risk and guide their decisions during evacuation. People often rely on the actions of others in uncertain situations, leading to patterns of herding behaviour, where individuals follow the majority, driven more by group dynamics than individual assessment (Huang, Lindell et al. 2012, Becker, Taylor et al. 2015, Petrucci 2022, Wang, Zhuang et al. 2024, Zhang, Liu et al. 2024). To build accurate, predictive models of human response during flood emergencies, robust empirical data are needed, yet such data are difficult to obtain due to the hazardous and unpredictable nature of real-world flood events.

To address this gap, this study employed Virtual Reality (VR) technology to simulate immersive, controlled flood evacuation scenarios that approximate real-world conditions while safely capturing human behavioural responses. By investigating human behaviour under the influence of crowd dynamics during flood evacuation, this research aligns with the scope of *Natural Hazards and Earth System Sciences* by advancing the understanding of human and societal factors in the monitoring and modelling of flood as a natural hazards.

VR has proven effective in emergency research, particularly fire evacuation, by enabling detailed, repeatable observation of behaviour under high-risk conditions without endangering participants (Lawson and Burnett 2015, Kinateder and Warren 2016, Deb, Carruth et al. 2017, Liu and Liu 2025). It allows for systematic manipulation of variables such as hazard severity and social cues (Kinateder, Müller et al. 2014, Shaw, Roper et al. 2019). Though still emerging in flood studies (e.g., (Fujimi and Fujimura 2020, Mol, Botzen et al. 2022, Simpson, Padilla et al. 2022, D'Amico, Bernardini et al. 2023, Denda and Fujikane 2024, Aksa, Ashar et al. 2025)), VR offers a promising, non-invasive method for generating empirical data on human behaviour in flood scenarios.

Studying the influence of social cues, particularly crowd behaviour on human decision-making is the primary focus of this research. This factor was incorporated into four VR experiments, while key environmental variables including floodwater level, destination clarity, and crowd size were systematically varied. There is limited knowledge on how each of these factors individually influences human decision-making during flood evacuation, which justifies examining them separately in a controlled, stepwise manner. In addition, given that measuring multiple complex decision factors and their interactions within a single VR scenario could overload participants and complicate result interpretation, the experiments were structured as a stepwise sequence, with each study building on the findings of the previous one and the outcomes reported in the literature. VR1 as a feasibility study, examined the baseline effect of crowd behaviour on route choice; VR2 explored how variations in crowd size influence this effect; VR3 investigated how destination clarity modulates reliance on social cues; and VR4 assessed how different floodwater levels alter the impact of crowd behaviour. Together, these controlled, interrelated studies provide a sequential and coherent examination of how social and environmental factors interact to shape human behaviour in flood evacuation scenarios.

# 2 Methods

# 2.1 Participants

A total of 84 participants (50% male, 50% female) took part in the four within-subject VR experiments, with the majority aged 20–30 (69%). Recruitment was conducted through emails, posters, and flyers distributed among students and staff at the University of Nottingham.

VR1 was conducted as feasibility study with minimum sample size to guide the rest of the research. A power analysis (G\*Power 3.1.9.7 (Faul, Erdfelder et al. 2007)) for a one-way repeated measures ANOVA with three conditions (f = 0.40,  $\alpha = .05$ , power = .80) indicated a required minimum sample of N = 12 for VR1. For VR2-VR4, each with four conditions, a power analysis using a smaller effect size (f = .25,  $\alpha = .05$ , power = .80) indicated a required sample of N = 24 per study. In each VR study, all participants completed all experimental conditions (VR1: 3 trials; VR2-VR4: 4 trials each) in a counterbalanced order to control for order effects. These sample sizes were sufficient to detect medium-to-large effects (Cohen 2013) while providing reliable and generalisable insights into human decision-making under varying crowd and environmental conditions.

# 0 2.2 Unity3D Setup

The virtual environment was developed in Unity3D, a widely used game engine for immersive simulations. It is specifically used to model a realistic flood road scenario across four experimental conditions (Figure 1).

110

120

#### 2.2.1 Materials

The model comprised three main components: the player, non-player characters (NPCs), and the environment, which included flooded roads, vehicles, and infrastructure. To enhance realism, the environment was augmented with rain effects and ambient sounds (e.g., heavy rain, helicopters, ambulance sirens). Animated NPCs sourced from the Unity Asset Store ((n.d.)) and Sketchfab (Sketchfab) represented crowds exhibiting either safe or risky behaviours, serving as social cues under different experimental conditions. Additional development tools included EasyRoad3D for road layout, Unity's terrain system and UK-specific assets for contextual realism, and a water shader from the Unity Asset Store to simulate floodwater.

The Unity3D scene was implemented with the three components, player, NPCs, and environment, whose movements and interactions were modelled as follows.

# 2.2.2 Components Setup

**Player:** Dynamic colliders with rigid bodies were assigned to the player, allowing it to respond to physics (gravity, forces, collisions), and to move, rotate, and interact with other colliders.

**NPCs:** NPCs were also assigned dynamic colliders with rigid-bodies, enabling them to respond to physics and interact mechanically with the environment. Their movement was governed by predefined destinations, which they approached either via the shortest path or through intermediate waypoints, depending on the experimental condition.

**Environment (solid objects):** Static colliders (without rigid-bodies) were applied to immovable objects such as cars, road elements, trees, and buildings. These colliders ensured that other components could collide with them, but the objects themselves remained unaffected by forces.

**Flood water:** Flood water was represented using static shaders simulating the water surface, with animated wave and current effects. These served as visual rather than mechanical components of the environment.

# 2.2.3 Interaction Modelling

Player-NPCs: Both the player and NPCs had dynamic colliders with rigid-bodies, preventing them from passing through
each other. No psychological interactions were assigned to NPCs in response to the player; instead, interactions were driven by the participant's perception of the NPCs.

Player/NPCs (solid objects): A navigation mesh was defined over the terrain (including lower-height objects) to guide the movement of the player and NPCs. Additionally, static colliders prevented passing through solid objects. No psychological forces were defined for NPCs in this context, making interactions purely mechanical. For the player, interactions with solid objects were influenced both by the participant's perception of the objects and by the mechanical forces exerted through colliders.

**NPC-NPC:** Both mechanical and psychological interactions were modelled. Mechanical forces arose from the dynamic colliders, while psychological forces followed Helbing's social force model (Helbing, Farkas et al. 2002), prompting NPCs to adjust their trajectories to maintain a certain minimum distance from each other.

Player-Flood Water: No mechanical interactions were implemented between the player and flood water. The interaction was entirely psychological, determined by the participant's perception of the situation and action.

**NPC-Flood Water:** NPC-water interactions were psychological and based on water depth. At each simulation step, if the water level within a defined radius around the NPC exceeded a scenario-specific threshold, the NPC altered its trajectory to avoid entering water deeper than that threshold, demonstrating safe crowd behaviour. In contrast, when this adjustment did not occur, NPCs passed through the floodwater, demonstrating risky crowd behaviour.

# 2.3 VR Configurations



The VR setup was deployed on a PICO 4 headset (XR) for VR1 and on a Meta Quest Pro (Meta) for VR2 through VR4.

Figure 1. Experiment VR scene with scenario details (SP = Start Point, RR=Risky Route, SR=Safe Route, SD=Safe Destination).

To support clarity across experimental conditions, the following terms are defined:

**Safe Destination:** The designated end point where the rescue team is located, including an ambulance, an emergency helicopter, and several NPCs in yellow vests as rescue team.

**Crowd:** NPCs around participants at the start point take the route to the safe destination based on the experimental condition. The crowd behaviours are categorized as Risky and Safe.

Risky Route: The direct path to the safe destination that requires participants to cross the floodwater.

Safe Route: The alternate path leading to the same destination via a hilly route and over a bridge, allowing to avoid floodwater.

**Large Crowd:** The group of twenty NPCs departing from the starting point toward the safe destination.

Small Crowd: The group of five NPCs taking the same route from the start point (Figure 2).

**Visible/Known Safe Destination:** The condition in which the safe destination is clearly visible to participants from the starting point, located across the flooded road and elevated above the bridge.

Invisible/Unknown Safe Destination: A condition in which participants start the experiment without seeing the safe destination and need to find it. The designated safe destination is at the end of the road over the bridge, on the opposite side of the experiment's starting point.

**High Water Level:** Water depth reaches approximately chest to shoulder height of NPCs and rises above the windows of nearby flooded vehicles (Figure 3).

**Low Water Level:** Water depth is shallower than previous level, reaching ankle height of NPCs and the base of the cars' wheels.

**Pre-test Simulation:** the simplified version of the VR scene (without floodwater, vehicles, crowd, or rescue elements) used for participant familiarisation.

Participants completed a series of standard and custom questionnaires at key stages of the experiment, including a demographic survey, the Simulator Sickness Questionnaire (Kennedy, Lane et al. 1993), the Igroup Presence Questionnaire (IPQ) (Schubert 2003), and a brief Likert-scale decision-making questionnaire assessing the influence of environmental and social factors on route choice. Qualitative data were also collected through post-condition interviews.

#### 2.4 Procedure




This research employed a within-subjects design across all four VR studies (Table1), with conditions counterbalanced. After reading an information sheet, informed consent, and pre-experiment questionnaires, participants carried out a 2–3-minute pre-trial to familiarise themselves with the VR environment and controllers.

Figure 2. Experiment VR scenes demonstrating the large (left) and small crowds (Becker, Taylor et al.).

Figure 3. Experiment VR scenes demonstrating the high (left) and low (write) water levels.

To begin the experiment, participants received the following instruction:

"On a heavily rainy day, you were driving home when your car got stuck on a flooded road. You heard a warning message on the radio stating that there is a chance of a flash flood and that you need to evacuate the area and reach the designated safe destination. Once the experiment begins, you will find yourself outside your car, standing next to the flooded road. Your task is to navigate to the designated safe destination where the rescue team is located in the virtual environment."

After each condition, participants completed a brief questionnaire to rate the influence of decision factors on their path choices, and participated in an interview, repeating this process for all assigned conditions. At the end, they completed a final set of questionnaires. The overall procedure is illustrated in Figure 4.

Table 1. VR experiments (VR1 to VR4) aim and design.

| Experiment             | Aim of Study                                                                                                                                                                                                     | Experimental Conditions (Within-subject)                                                                                                                                                                               | N   |
|------------------------|------------------------------------------------------------------------------------------------------------------------------------------------------------------------------------------------------------------|------------------------------------------------------------------------------------------------------------------------------------------------------------------------------------------------------------------------|-----|
| Pre-Test<br>Simulation | To provide familiarity with the experiment environment to participants and learn how to navigate around the scene with controllers                                                                               | Carried out in advance of all VR1 to VR4 experiments                                                                                                                                                                   | All |
| VR1                    | To understand how the presence of a crowd exhibiting safe and risky behaviour can influence human decision making in choosing their route to a safe destination.                                                 | 1A: Risky Behaviour of Crowd<br>1B: Safe Behaviour of Crowd<br>1C: No Crowd                                                                                                                                            | 12  |
| VR2                    | To understand how the size of the crowd including small<br>and large exhibiting safe or risky behaviour, can<br>influence human decision making in choosing their route<br>to a safe destination.                | 2A: Risky Behaviour of Crowd + Very Large Crowd<br>2B: Risky Behaviour of Crowd + Small Crowd<br>2C: Safe Behaviour of Crowd + Very Large Crowd<br>2D: Safe Behaviour of Crowd + Small Crowd                           | 24  |
| VR3                    | To understand how visibility and invisibility of the safe destination in presence of a crowd exhibiting safe and risky behaviour, can influence human decision making in choosing route to the safe destination. | 3A: Risky Behaviour of Crowd + Visible Destination<br>3B: Risky Behaviour of Crowd + Invisible Destination<br>3C: Safe Behaviour of Crowd + Visible Destination<br>3D: Safe Behaviour of Crowd + Invisible Destination | 24  |
| VR4                    | To understand how different level of flood water including low and high, in presence of a crowd can influence human decision making in choosing their route to the safe destination.                             | 4A: Risky Behaviour of Crowd + Low Water Level<br>4B: Risky Behaviour of Crowd + High Water Level<br>4C: Safe Behaviour of Crowd + Low Water Level<br>4D: Safe Behaviour of Crowd + High Water Level                   | 24  |



#### 3 VR1: Crowd Behaviour

#### **185 3.1 Rationale**





Past research has demonstrated that during natural disasters and emergency evacuations, individuals' decision-making is significantly influenced by the behaviour of those around them. The observation of crowd behaviour plays a crucial role in shaping

individuals' perceptions of the situation and guiding their subsequent actions. This influence often manifests in behaviours such as following others or engaging in herding dynamics (Helbing, Farkas et al. 2002, Huang, Lindell et al. 2012, Becker, Taylor et al. 2015, Petrucci 2022, Wang, Zhuang et al. 2024, Zhang, Liu et al. 2024). Referred to as *social cues*, these subtle behavioural signals affect how people interpret events and respond to crises (Wang, Zhuang et al. 2024). Gaining a deeper understanding of the mechanisms and dynamics of human behaviour under the influence of social cues in flood emergencies is essential for developing accurate and realistic behavioural models. In response to this need, as a feasibility study, VR1 experiment was designed to examine the influence of crowd behaviour on individual decision-making during a simulated flood evacuation scenario. It served as a feasibility study to inform the subsequent research and to identify additional factors that could be systematically integrated with crowd behaviour.

# 3.2 Results

#### 200 **3.2.1 Chosen Path**

Participants' path choices across the three conditions are summarised in Table 2. In both the Safe (1A) and Control (1C) conditions, 11/12 participants (91.7%) chose the safe route, while this dropped to 7/12 (58.3%) in the Risky (1B) condition, suggesting that observing risky crowd behaviour prompted a shift toward riskier decisions.

Cochran's Q test revealed a significant difference in decision patterns across conditions (p = 0.04), indicating that crowd behaviour influenced evacuation choices. Post hoc McNemar's tests with Bonferroni adjustment ( $\alpha = 0.016$ ) showed no significant pairwise differences (p > 0.016), likely due to limited sample size (Field 2024). Nevertheless, the trend is clear: participants were over four times more likely to select the risky route when exposed to risky crowd behaviour. This supports prior evidence of herding effects, where individuals follow the observable actions of others in uncertain or hazardous environments (Helbing, Farkas et al. 2002, Petrucci 2022, Wang, Zhuang et al. 2024), even when those actions involve crossing dangerous floodwaters (Fujimi and Fujimura 2020).

#### 3.2.2 Pre-movement Time

Pre-movement time, also known as response time or pre-evacuation time, refers to the interval between a stimulus and the start of action (Adrian, Amos et al. 2025). In this study, it is defined as the time from the simulated flood warning to the





participant's initiation of evacuation. In the VR environment, this was measured as the time spent observing and assessing before moving to the safe destination, extracted from VR screen recordings.

Participants in the Risky condition exhibited the longest decision-making time (1B; M = 13.8 s, SD = 4.4), compared to the Safe condition (1A; M = 7.5 s, SD = 4.5) and Control condition (1C; M = 7.08 s, SD = 2.4) (Table 2). A one-way repeated measures ANOVA revealed a significant effect of condition on pre-movement time, F(1.46, 16.06) = 11.81, p = .001. Post-hoc comparisons with Bonferroni correction showed that pre-movement time in the Risky condition was significantly longer than both the Safe (p = .025) and Control conditions (p = .001), while no significant difference was found between Safe and Control (p = 1.0).

These results suggest that observing a crowd engaging in risky behaviour increased participants' deliberation time before acting, possibly due to heightened uncertainty, cognitive conflict, or increased risk appraisal. This aligns with the notion that social cues influence not only the direction of movement but also the latency of evacuation decisions (Bode and Codling 2019).

#### 3.2.3 Decision Factors

Participants rated how different factors influenced their route choices (Table 3). The presence of the crowd and the crowd's choice of path were rated moderately to highly influential, with no significant differences between conditions 1A and 1B (p = 0.95 and p = 0.46 respectively). Similarly, both the overall flood water condition and the flood water level were consistently rated as highly influential across conditions 1A, 1B, and 1C, although no significant differences were found (p = 0.60 and p = 0.97, respectively). These findings suggest that while participants acknowledged the importance of all four factors in their route choice decisions, the experimental variations did not significantly alter their perceived influence.

Qualitative data revealed varied perceptions of flood severity. While some participants saw the water as highly dangerous, describing it as shoulder-deep or submerging cars, others downplayed it as "a little high" or "under the waist." Misinterpretations, including mistaking the river for the road, indicated more uncertainty. Concerns about depth, flow, and hidden debris also influenced avoidance. Responses ranged from complete rejection ("I completely rejected that route") to confidence ("It's easy to cross"). Decisions were also influenced by perceptions of flood severity, including submerged vehicles, unpredictable flow, and risks to personal belongings, reflected in high quantitative ratings of flood conditions across all scenarios.

The crowd had a mixed influence: some found reassurance in following others, while others were indifferent, avoided the crowd, or remained unaware of it, focusing solely on reaching safety. Risk-taking was often influenced by observing others using the same route, with participants citing crowd behaviour as a cue for safety. In contrast, absence of a crowd led some to make independent and occasionally riskier choices due to a lack of social cues. Additional factors influencing decisions included path clarity, distance, environmental sounds (e.g. sirens), and obstacles like moving cars or slippery terrain. Some prioritized safety and chose longer, clearer routes, while others favoured the directness of the flooded path. In rare cases, moral or social responsibility, such as helping vulnerable individuals or seeking group support, also shaped choices.

Overall, while the quantitative data showed consistent influence of all four main decision factors across conditions, the qualitative data revealed the nuanced and often conflicting personal interpretations that shaped individual decision-making during the simulated flood scenarios.

# 4 VR2: Crowd Behaviour and Crowd Size

#### 4.1 Rationale



Building on VR1, which demonstrated that crowd behaviour influences flood evacuation decisions, VR2 was designed to examine how crowd size modulates this effect, directly guided by the patterns observed in VR1. In VR1, while participants frequently reported perceiving the crowd size as large, the study confirmed herding tendencies and showed that individuals responded differently to the large crowd size, which directly informed the choice of experimental conditions in VR2.

Research shows that group size shapes evacuation dynamics by increasing conformity and perceived legitimacy and trust (Haghani, Sarvi et al. 2019, Kinateder and Warren 2021). Yet, large crowds can also heighten anxiety, signal congestion or risk, and prompt avoidance, especially in ambiguous, high-risk settings. Most of this work, however, focuses on indoor or fire-related scenarios, with limited application to floods, which involve complex and visually deceptive hazards.

VR1 also indicated that participants used crowd presence as a safety heuristic but interpreted their feeling about the large size of crowd differently depending on personal risk perception and environmental appraisal. Importantly, prior studies have not isolated the specific impact of crowd size from other factors like hazard severity or path clarity. Therefore, VR2 aims to explore how variations in crowd size influence individual evacuation decisions in a flood scenario by manipulating crowd size and their behaviour.

#### 265 **4.2 Results**



#### 4.2.1 Chosen Path

Safe route selection was highest in Safe-Small (2A) and Safe-Large (2C) conditions (95.8%), and notably lower in Risky-Small (2B, 79.2%) and Risky-Large (2D, 70.8%). This suggests that risky crowd behaviour decreased safe decisions, particularly when group size was large. These findings indicate that risky crowd behaviour reduced safe choices, especially with larger crowds.

Cochran's Q test revealed a significant effect of condition on evacuation choices (p = 0.006), however post hoc McNemar tests with Bonferroni correction ( $\alpha = 0.0083$ ) showed no significant pairwise differences, Overall, VR2 findings highlight the influence of social cues on evacuation decisions. Participants were more likely to choose the risky path when in larger groups exhibiting risky behaviour. However, no significant difference emerged between any individual condition pairs, suggesting that the lack of significance failed to provide evidence for an influence of crowd size on chosen path.

Table 2. VR1 to VR4 results on choice of path and pre-movement time (significance \*)

|          | Independent Variable |                    |         |                         | Participants Responses- Choice of Path |                           |                                  | Participants Pre-movement Time             |                                                                             |                                                     |                                             |                                               |                                     |                                              |                                              |                 |                                              |  |
|----------|----------------------|--------------------|---------|-------------------------|----------------------------------------|---------------------------|----------------------------------|--------------------------------------------|-----------------------------------------------------------------------------|-----------------------------------------------------|---------------------------------------------|-----------------------------------------------|-------------------------------------|----------------------------------------------|----------------------------------------------|-----------------|----------------------------------------------|--|
| VR Study | Condition            | Level1             | Value   | Level 2                 | Value                                  | (Prol                     | ponse<br>bability<br>%)<br>Risky | Cochran<br>'s Q Test                       | Post Hoc<br>Pairwise<br>Comparisons:<br>McNemar Test                        | Mea<br>n<br>(s)                                     | SD                                          | One-Way<br>Repeated<br>Measures<br>ANOVA Test | Post Hoc<br>Pairwise<br>Comparisons | Commen<br>t                                  |                                              |                 |                                              |  |
|          | 1A                   |                    | Safe    |                         | N/A                                    | 11/12                     | 1/12                             | Г                                          |                                                                             | 7.5                                                 | 4.5                                         |                                               | F                                   |                                              |                                              |                 |                                              |  |
|          | IA                   | 0                  |         |                         | IN/A                                   | (91.7) (8.3)              | 1A vs 1B = 0.125                 | 1.5                                        | 4.5                                                                         | F (1.4.16.06)                                       | 1A vs 1B =.025                              |                                               |                                     |                                              |                                              |                 |                                              |  |
| VR1      | 1B                   | Crowd<br>Behaviour | Risky   | N.A.                    | N/A                                    | (58.3)                    |                                  | 0.04* 1A vs 1C = 0.100<br>1B vs 1C = 0.125 | 13.8                                                                        | 4.4                                                 | =11.8                                       | 1A vs 1C = 1.0                                | -                                   |                                              |                                              |                 |                                              |  |
|          | 1C                   |                    | Control |                         | N/A                                    | 11/12<br>(91.7)           | 1/12<br>(8.3)                    |                                            | 1B VS 1C = 0.125                                                            | 7.08                                                | 2.4                                         | p = 0.001*                                    | 1B vs 1C = <b>0.001</b> *           |                                              |                                              |                 |                                              |  |
|          | 2A                   |                    | Safe    |                         | Small                                  | 23/24<br>(95.8)           | 1/24<br>(4.2)                    |                                            | 2A vs 2B = 0.125                                                            | 8.6                                                 | 5.2                                         |                                               | 2A vs 2B = <b>0.001</b> *           |                                              |                                              |                 |                                              |  |
| VR2      | 2B Cro               | Crowd              | Risky   | Crowd Size              | Small                                  | 19/24<br>(79.2)           | 5/24<br>(20.8)                   | B) 0.000*                                  | 0.006* 2A vs 2D = 0.0<br>2B vs 2C = 0.0<br>2B vs 2D = 0.6<br>2B vs 2D = 0.6 | .8)<br>24<br>2) <b>0.006*</b>                       | 2B vs 2C = 0.125<br>2B vs 2D = 0.62         | 0.000*                                        | 2A vs 2D = 0.031                    | 13.5                                         | 5.1                                          | F (3,69) = 17.8 | 2A vs 2C = 0.71<br>2A vs 2D = <b>0.001</b> * |  |
| VKZ      | 2C                   | Behaviour          | Safe    | Crowd Size              | Large                                  | (95.8) (4.2)              | 1/24<br>(4.2)                    | 95.8) (4.2)                                |                                                                             |                                                     |                                             | 8.3                                           | 5.1                                 | p = <b>0.001</b> *                           | 2B vs 2C = <b>0.001</b> *<br>2B vs 2D = 0.14 |                 |                                              |  |
|          | 2D                   |                    | Risky   |                         | Large                                  | 17/24<br>(70.8)           | 7/24<br>(29.2)                   |                                            | 2C vs 2D = 0.03                                                             | 15.4                                                | 7.3                                         |                                               | 2C vs 2D = 0.001*                   |                                              |                                              |                 |                                              |  |
|          | 3A                   |                    | Safe    |                         | Known                                  | 24/24<br>(100)            | 0/24<br>(0.0)                    |                                            | 3A vs 2B = 0.08<br>3A vs 2C = N. A                                          | 10.2                                                | 3.6                                         |                                               | 3A vs 3B = 0.05<br>3A vs 3C =       |                                              |                                              |                 |                                              |  |
|          | 3B                   | Risk               | Risky   | Clarity on              | Known                                  | 15/24<br>(62.5)           | 9/24<br>(37.5)                   |                                            | 3A vs 2D = 0.001*                                                           | 0.001*                                              | 8.1                                         | 2.7                                           |                                     | <b>0.001*</b><br>3A vs 3D =                  | Two                                          |                 |                                              |  |
| VR3      | 3C                   | Crowd<br>Behaviour | Safe    | Location of<br>the Safe | Unknown                                | 24/24<br>(100)            | (0.00)<br>4 12/24<br>(50)        | 0/24<br>(0.00) <b>0.001</b> *              | 3B vs 2C = 0.008*                                                           | 11.9                                                | 2.9                                         | F (3,63) = 18.4<br>p = <b>0.001</b> *         | <b>0.001*</b><br>3B vs 3C = 0.18    | premovem<br>ent time is                      |                                              |                 |                                              |  |
|          | 3D                   |                    | Risky   | Destination             | Unknown                                | 12/24<br>(50)             |                                  |                                            |                                                                             | 3B vs 2D =<br>0.125<br>3C vs 2D =<br><b>0.001</b> * | 14.9                                        | 4.3                                           |                                     | 3B vs 3D =<br>0.001*<br>3C vs 3D =<br>0.001* | missed.                                      |                 |                                              |  |
|          | 4A                   |                    | Safe    |                         | High                                   | 24/24 0/24<br>(100) (0.0) |                                  |                                            |                                                                             | 4A vs 2B = 0.001*                                   | 8.3                                         | 4.5                                           |                                     |                                              |                                              |                 |                                              |  |
|          | 4B                   | Crowd              | Risky   | Flood Water             | High                                   | 19/24<br>(79.1)           | (20.8)                           | 0.001*                                     |                                                                             | (20.8)<br>5/24 <b>0.001</b> *                       | 4A vs 2C = <b>0.001</b> *<br>4A vs 2D =0.21 | 9.4                                           | 4.6                                 | F (3,66) =14.8                               |                                              |                 |                                              |  |
| VR4      | 4C                   | Behaviour          | Safe    | Level                   | Low                                    | 20/24<br>(83.3)           |                                  |                                            |                                                                             |                                                     | 0/24                                        | 4B vs 2C = N.A<br>4B vs 2D =                  | 7.9                                 | 3.9                                          | p = 0.28                                     | -               | -                                            |  |
|          | 4D                   |                    | Risky   |                         | Low                                    | 8/24<br>(33.3)            | 16/24<br>(66.6)                  |                                            | 0.008*<br>4C vs 2D =<br>0.008*                                              | 9.5                                                 | 4.8                                         |                                               |                                     |                                              |                                              |                 |                                              |  |

Table 3: VR1 Decision Factors Questionnaire Results

| <b>Decision Factor</b>        | Condition | Mean<br>(SD) | Median | Test                 | P    | Post-Hoc<br>Comparisons |  |
|-------------------------------|-----------|--------------|--------|----------------------|------|-------------------------|--|
|                               | 1A        | 3.5 (1.37)   | 4      |                      | 0.95 | -                       |  |
| Presence of the crowd         | 1B        | 3.7 (1.6)    | 4.5    | Wilcoxon Signed-Rank |      |                         |  |
|                               | -         | 1            | ı      |                      |      |                         |  |
|                               | 1A        | 3.1 (1.1)    | 3      |                      |      | -                       |  |
| Crowd choice of path          | 1B        | 3.6 (1.5)    | 4      | Wilcoxon Signed-Rank | 0.46 |                         |  |
| _                             | -         | -            | -      | _                    |      |                         |  |
| Fl d4 II                      | 1A        | 4.42 (1.1)   | 5      |                      | 0.60 | -                       |  |
| Flood water overall condition | 1B        | 4.0 (1.4)    | 5      | Friedman's ANOVA     |      |                         |  |
| condition                     | 1C        | 4.7 (0.4)    | 5      |                      |      |                         |  |
|                               | 1A        | 3.7 (1.7)    | 4.5    |                      | 0.97 |                         |  |
| Flood water level             | 1B        | 4.8 (0.9)    | 4      | Friedman's ANOVA     |      | -                       |  |
|                               | 1C        | 3.8 (1.4)    | 4.5    |                      |      |                         |  |

# 4.2.2 Pre-movement Time

Participants in Risky-Large (2D) exhibited the longest pre-movement time (M = 15.4 s, SD = 7.3), followed by Risky-Small (2B; M = 13.5 s, SD = 5.1). In contrast, Safe-Small (2A; M = 8.6 s, SD = 5.2) and Safe-Large (2C; M = 8.3 s, SD = 5.1) conditions were associated with shorter decision times. A one-way repeated measures ANOVA revealed a significant effect of condition on pre-movement time in VR2, F (3, 69) = 17.8, p = 0.001. Decision-making times varied notably with both crowd behaviour (Safe vs. Risky) and crowd size (Small vs. Large).







Post hoc comparisons with Bonferroni correction (α = 0.0083) showed that Risky-Small (2B) differed significantly from 290 Safe-Small (2A) and Safe-Large (2C), and Risky-Large (2D) differed significantly from Risky-Small (2B), demonstrating significantly longer pre-movement times. However, no significant differences were found between the two risky conditions or between the two safe condition, as these did not meet the Bonferroni-adjusted threshold.

In agreement with VR1 results, these findings indicate that risky crowd behaviour, irrespective of the size of the group, delayed participants' initiation of movement. The results reinforce the idea that social cues not only shape the direction of evacuation decisions but also affect the speed with which individuals choose to act in emergencies.

#### 4.2.3 Decision Factors

Participants' route choices were shaped by a combination of social and environmental factors across the four experimental conditions. Table 4 summarizes the median ratings for the key variables influencing their decision-making. A Friedman test revealed significant effects for four social variables, presence of the crowd, path choice, crowd size, and trust (p 






In summary, participants' route choices were shaped more by social cues than by actual environmental conditions. Trust in the crowd and observed behaviour significantly influenced decisions, with larger crowds amplifying these effects (however not significantly in Risky conditions being seen as either reassuring or overwhelming depending on context. Despite uniform flood hazards, subjective risk perceptions varied, driven by group dynamics, emotional responses, and ambient cues rather than objective danger.

In summary, participants' route choices were shaped more by social cues than by actual environmental conditions. Trust in the crowd and observed behaviour significantly influenced decisions, with larger crowds amplifying these effects (however not significantly in Risky conditions) being seen as either reassuring or overwhelming depending on context. Despite uniform flood hazards, subjective risk perceptions varied, driven by group dynamics, emotional responses, and ambient cues rather than objective danger.

# 5 VR3: Crowd Behaviour and Clarity on Safe Destination

#### 5.1 Rationale

Building on the findings of VR1 and VR2, this study (VR3) examines how the clarity and visibility of a safe evacuation destination influenced decision-making during floods in the presence of social cues. Leading on from the VR2 finding that participants reported on the influence of knowledge of destination, earlier studies similarly found that a clear, visible, and reachable safe destination strongly guided participants' path choice. Literature supports that spatial knowledge and destination visibility significantly influence path choices. When desired destinations are not visible, individuals rely on external cues like signage (Gärling, Böök et al. 1986) or, in our flood scenarios, social cues. A lack of knowledge about evacuation targets can increase flood risk (Sadeghi-Pouya, Nouri et al. 2017). Fire evacuation research also shows that visible exits are more likely to be used (Haghani and Sarvi 2017, Fu, Liu et al. 2024). Simulation studies further suggest that social settings and prior knowledge of a space (e.g., exits) shape crowd behaviour (Chu, Parigi et al. 2015). Given these insights and prior findings, VR3 investigates how clarity on location of a safe destination affects evacuation decisions under different crowd behaviours.

# **345 5.2 Results**

#### 5.2.1 Chosen Path

Under varying levels of clarity regarding the location of the safe destination, Cochran's Q test was conducted to determine whether the proportion of participants choosing the safe versus risky path differed significantly across the four VR3 conditions. The test revealed a statistically significant difference in participants' decision patterns (p = 0.001), indicating that both crowd behaviour and destination clarity influenced path choice. However, post hoc pairwise comparisons using McNemar's test with Bonferroni correction showed no significant difference between Risky conditions (3B and 3D),


suggesting that the lack of destination clarity did not exert a stronger effect on risky response when participants were exposed to risky crowd behaviour. No significant difference in path choices was observed between conditions 3A and 3C as well.

These findings suggest that while crowd behaviour influences evacuation choices, the effect of destination clarity on selecting the safe route remains limited, particularly when individuals are exposed to risky crowd cues.

# 5.2.2 Pre-movement Time

A one-way repeated measures ANOVA revealed a significant effect of condition on pre-movement time in VR3, F (3,63) = 18.4, p = .001. Pre-movement times varied across the four experimental conditions that manipulated both the observed crowd behaviour (Safe vs. Risky) and the clarity of the safe destination (Known vs. Unknown).

Post hoc comparisons with Bonferroni correction showed that participants in the Risky-Unknown (3D) exhibited the longest pre-movement time (M = 14.9 s, SD = 4.3), which was significantly longer than all other conditions. Safe-Unknown (3C) also led to significantly longer decision times (M = 11.9 s, SD = 2.9) compared to known conditions. The shorter pre-movement time were observed in known conditions.

Table 4. VR2 Decision Factors Questionnaire Results (significance \*)

| Decision Factor   | Condition    | Mean (SD)   | Median | Friedman<br>Anova test | Post-Hoc Comparisons<br>Wilcoxon Signed-Rank<br>test |
|-------------------|--------------|-------------|--------|------------------------|------------------------------------------------------|
|                   | 2A           | 2.83 (1.78) | 2.5    |                        | 2A  vs  2B = 0.018                                   |
|                   | 2B           | 2.04 (1.60) | 1      |                        | 2A  vs  2C = 0.061                                   |
| Presence of the   | 2C           | 3.63 (1.58) | 4.5    | 0.003*                 | 2A  vs  2D = 0.97                                    |
| crowd             |              | , ,         |        | 0.005                  | 2B  vs  2C = 0.001*                                  |
|                   | 2D           | 2.79 (1.56) | 2.5    |                        | 2B  vs  2D = 0.014                                   |
|                   |              |             |        |                        | 2C  vs  2D = 0.047                                   |
|                   | 2A           | 2.7 (1.69)  | 2.5    |                        | 2A  vs  2B = 0.07                                    |
|                   | 2B           | 2.0 (1.66)  | 1      |                        | 2A  vs  2C = 0.012                                   |
| Crowd choice of   | 2C           | 3.7 (1.6)   | 4.5    | 0.001*                 | 2A  vs  2D = 0.388                                   |
| path              |              |             |        | 0.001                  | 2B  vs  2C = 0.003*                                  |
|                   | 2D           | 2.4 (1.7)   | 1      |                        | 2B  vs  2D = 0.41                                    |
|                   |              |             |        |                        | 2C  vs  2D = 0.021                                   |
|                   | 2A           | 2.4 (1.5)   | 1.5    |                        | 2A  vs  2B = 0.19                                    |
|                   | 2B           | 1.8 (1.3)   | 1      | 0.008*                 | 2A  vs  2C = 0.036                                   |
| Size of the crowd | 2C           | 3.2 (1.5)   | 3.5    |                        | 2A  vs  2D = 0.87                                    |
| Size of the crowd |              |             | 2      |                        | 2B  vs  2C = 0.013*                                  |
|                   | 2D           | 2.5 (1.5)   |        |                        | 2B  vs  2D = 0.108                                   |
|                   |              |             |        |                        | 2C vs 2D =0.093                                      |
|                   | 2A           | 3.5 (1.4)   | 4      |                        |                                                      |
| Flood water       | 2B           | 3.8 (1.3)   | 4      | 0.34                   |                                                      |
| overall condition | 2C           | 3.6 (1.6)   | 4.5    | 0.54                   | -                                                    |
|                   | 2D           | 3.3 (1.4)   | 4      |                        |                                                      |
|                   | 2A           | 3.5 (1.3)   | 4      |                        |                                                      |
| Flood water level | 2B           | 3.8 (1.4)   | 4      | 0.32                   |                                                      |
| rioou water level | 2C           | 3.8 (1.5)   | 5      | 0.32                   | -                                                    |
|                   | 2D           | 3.6 (1.5)   | 4      |                        |                                                      |
|                   | 2A           | 2.5 (1.3)   | 2      |                        | 2A  vs  2B = 0.034                                   |
|                   | 2B           | 1.7 (1.2)   | 0      |                        | 2A  vs  2C = 0.086                                   |
| Trust on the      | 2C           | 3.1 (1.7)   | 3.5    | 0.01*                  | 2A  vs  2D = 0.190                                   |
| crowd             |              | ` /         |        | - 0.01*                | 2B  vs  2C = 0.008*                                  |
|                   | 2D 2.0 (1.5) | 2.0 (1.5)   | 1      |                        | 2B  vs  2D = 0.47                                    |
|                   |              |             |        |                        | 2C  vs  2D = 0.021                                   |







These findings indicate that both crowd behaviour and destination clarity significantly affect evacuation latency. The longest delays occurred under risky social cues combined with spatial uncertainty, suggesting increased cognitive load and hesitation. In contrast, the fastest decisions followed safe cues and clear destinations. Alongside VR1 and VR2, these results underscore that social and environmental factors jointly shape not only evacuation choices but also the speed of decision-making in emergencies.

#### 5.2.3 Decision Factors

As shown in Table 5, presence of the crowd and trust in the crowd were rated as significantly more influential than other factors (p = 0.0117 and p = 0.033, respectively). However, post hoc Wilcoxon paired comparisons with Bonferroni correction revealed no statistically significant pairwise differences. Ratings for floodwater condition and water level remained consistently high across all conditions but did not differ significantly (p > .05), indicating uniform perception of environmental risk regardless of scenario variation.

Qualitative data supported the quantitative findings by highlighting the dominant role of social context in shaping route choices. In conditions with safe crowd behaviour (3A, 3C), participants described the crowd as reassuring and often followed their direction without hesitation, especially in unfamiliar settings: "I felt safer because everyone was heading the same way, even if I didn't know where I was going." In contrast, in risky crowd scenarios (3B, 3D), participants were more divided. Some followed the crowd out of urgency or perceived trust while unclear on the location of the safe destination, while others chose independent paths, expressing mistrust or relying on prior knowledge: "I didn't trust them, they were heading into the flood."

In agreement with VR1 and VR results, despite identical environmental setups, perceptions of floodwater level and danger varied. Participants often inferred risk based on crowd behaviour, with several noting the water seemed less threatening when others crossed it: "It didn't look too deep since they went through." Others described the water as hazardous, citing hidden debris, electricity, or strong currents: "Even if it looked shallow, I wasn't going to risk it." These findings point to socially modulated hazard perception.

Prior experience played a critical role, particularly in conditions with a known destination (3A, 3B). Participants with earlier exposure to the environment reported greater confidence and chose safer routes more readily. Several stated that had the risky crowd and unknown destination (3D) appeared earlier in their sequence, they might have followed the crowd: "If this was my first condition, I probably would've gone with them." The identification of high ground or familiar landmarks further anchored decision-making.

Additional influences included auditory cues such as sirens and helicopter sounds, visual signals like light or vehicle movement, and emotional states such as panic or hesitation. Some participants reported misinterpreting cues, such as assuming police presence indicated a safe path: "I saw a policeman in the crowd, so I thought it must be the right way."

Others described following the crowd reflexively before realizing it led away from safety.


Overall, VR3 results reinforce patterns observed in prior studies, emphasizing the significant influence of crowd behaviour, particularly under conditions of uncertainty, on route choice decisions. While flood-related environmental factors remained consistently rated as important, they did not drive differential behaviour across scenarios. Instead, the combination of crowd behaviour and destination familiarity most strongly shaped both perceived influence and observed decisions during simulated flood evacuations.

Table 5. VR3 Decision Factors Questionnaire Results (significance \*)

| Decision Factor   | Condition | Mean<br>(SD) | Median | Friedman<br>Anova test<br>P | Post-Hoc Comparisons<br>Wilcoxon Signed-Rank test<br>P      |
|-------------------|-----------|--------------|--------|-----------------------------|-------------------------------------------------------------|
|                   | 3A        | 3.6 (1.12)   | 4      |                             | 3A  vs  3B = 0.09                                           |
| Presence of the   | 3B        | 3.1 (1.19)   | 3      | 0.0117*                     | 3A  vs  3C = 0.3<br>3A  vs  3D = 0.39                       |
| crowd             | 3C        | 3.9 (1.06)   | 4      |                             | 3B  vs  3C = 0.01<br>3B  vs  3D = 0.03                      |
|                   | 3D        | 3.8 (1.04)   | 4      |                             | 3C  vs  3D = 0.7                                            |
|                   | 3A        | 3.6 (1.2)    | 4      |                             |                                                             |
| Crowd choice of   | 3B        | 2.9 (1.4)    | 3      | 0. 25                       |                                                             |
| path              | 3C        | 3.8 (0.86)   | 4      |                             | -                                                           |
|                   | 3D        | 3.6 (1.1)    | 4      |                             |                                                             |
|                   | 3A        | 3.3 (1.3)    | 3.5    | 0.87                        |                                                             |
| Flood water       | 3B        | 3.3 (1.1)    | 3.5    |                             |                                                             |
| overall condition | 3C        | 3.2 (1.1)    | 3      | 0.67                        | -                                                           |
|                   | 3D        | 3.5 (1.06)   | 3.5    |                             |                                                             |
|                   | 3A        | 3.2 (1.3)    | 4      |                             |                                                             |
| Flood water level | 3B        | 3.2 (1.1)    | 3.5    | 0.52                        |                                                             |
| rioou water level | 3C        | 2.9 (1.2)    | 2.5    | 0.52                        | -                                                           |
|                   | 3D        | 3.3 (1.2)    | 4      |                             |                                                             |
|                   | 3A        | 3.7 (1.2)    | 4      |                             | 3A  vs  3B = 0.05                                           |
|                   | 3B        | 2.8 (1.4)    | 2.5    |                             | 3A  vs  3C = 0.63                                           |
| Trust on the      | 3C        | 3.5 (1.0)    | 4      | .033*                       | 3A  vs  3D = 0.24                                           |
| crowd             | 3D        | 3.3 (1.3)    | 3.5    | .033*                       | 3B  vs  3C = 0.06<br>3B  vs  3D = 0.09<br>3C  vs  3D = 0.39 |

# 6 VR4: Crowd Behaviour and Floodwater Level

# 405 6.1 Rationale


Building on previous studies, VR4 study examines how the physical characteristics of floodwater, particularly water level, influenced decision-making during floods in the presence of social cues. Previous VR studies showed that floodwater conditions, despite being constant across scenarios, were perceived differently depending on social context, suggesting that crowd behaviour modulates risk perception. This effect, seen when participants followed a risky crowd and perceived deep water as "doable," indicates a complex interplay between environmental appraisal and social influence.

VR4 builds on this by isolating the effect of water level under varied social conditions. Previous research has shown that water depth affects evacuation speed and walking stability (Bernardini, Quagliarini et al. 2020, Dias, Abd Rahman et al. 2021), and that increased water depth correlates with greater perceived risk and higher casualty rates (Arrighi, Oumeraci et al. 2017, Quagliarini, Romano et al. 2023). However, people often voluntarily step into floodwater when its characteristics

appear manageable, raising questions about which environmental thresholds alter behaviour (Becker, Taylor et al. 2015).

Moreover, familiarity with the environment, previously shown to reduce dependence on crowd cues, was also examined, as it may mitigate misperceptions in high-risk contexts (Fujimi and Fujimura 2020, Papagiannaki, Diakakis et al. 2021). Thus, VR4 aims to advance understanding of how floodwater affordance, perceived risk, and social dynamics interact to influence real-time evacuation choices.

#### 420 **6.2 Results**



#### 6.2.1 Chosen Path

To examine how crowd behaviour and floodwater level affected path choices, Cochran's Q test was conducted across the four VR4 conditions revealing a significant overall difference (p = 0.001).

Focusing on the two risky crowd conditions, participants were significantly less likely to choose the risky path in the Risky-425 High condition (4B), where none selected the risky option (0%), compared to the Risky-Low condition (4D), where 48% chose the risky path. This difference was statistically significant (p = 0.008, McNemar's test).

These results suggest that floodwater level moderated the influence of risky crowd behaviour. While risky crowd behaviour generally reduced safe path choices in earlier studies, in VR4, it was only effective in deterring risky decisions when paired with high water levels. In contrast, when floodwater was low, risky crowd behaviour no longer deterred participants from also choosing the risky path, indicating a boundary condition for the influence of social cues during evacuation.

# 6.2.2 Pre-movement Time

VR4 tested the impact of crowd behaviour and flood water level on pre-movement time. A one-way repeated measures ANOVA found no significant effect, F (3, 66) = 1.48, p = .28. Although Risky conditions (4B, 4D) showed slightly longer times (M = 9.4, 9.5 s) than Safe conditions (4A, 4C; M = 7.9, 8.3 s), these differences were not significant.

Unlike previous studies (VR1-VR3), neither social cues nor environmental risk alone significantly affected evacuation timing in this scenario, suggesting a possible reduced sensitivity to the variables tested.

#### **6.2.1 Decision Factors**

Quantitative results showed that flood water level was the most influential decision factor. A Friedman test revealed significant differences across conditions (p = .02), with condition 4B (risky behaviour high water) receiving the highest influence ratings. Post-hoc tests showed that 4B differed significantly from 4A (p = .044) and 4C (p = .008). In contrast, crowd-related factors presence, path choice, and trust did not vary significantly between conditions (p > .10), suggesting a steady, moderate influence (Table 6).

Qualitative interviews supported these findings, with many participants emphasizing concerns about high water levels, especially when the water reached chest height. Around 38% of participants mentioned their inability to swim as another key

reason for avoiding flooded routes. Perceptions of water level were more accurate when it was shallow, but as depth increased, participants increasingly under or overestimated the hazard, sometimes mistaking chest-level water for waist-deep. These mis judgments, often shaped by stress and poor visibility, influenced decisions even when actual conditions were identical.

**Table 6.** VR4 Decision Factors Questionnaire Results (significance \*)

| Decision Factor    | Condition | Mean (SD) | Median | Friedman<br>Anova test<br>P | Post-Hoc<br>Comparisons<br>Wilcoxon Signed-<br>Rank test<br>P  |
|--------------------|-----------|-----------|--------|-----------------------------|----------------------------------------------------------------|
|                    | 4A        | 3.7 (1.1) | 4      |                             |                                                                |
| Presence of the    | 4B        | 2.7 (1.5) | 2      | 0.14                        |                                                                |
| crowd              | 4C        | 3.0 (1.3) | 3      | 0.14                        | -                                                              |
|                    | 4D        | 3.2 (1.2) | 3      |                             |                                                                |
|                    | 4A        | 3.7 (1.3) | 4      |                             |                                                                |
| Crowd choice of    | 4B        | 2.8 (1.6) | 2      | 0.10                        |                                                                |
| path               | 4C        | 3.3 (1.4) | 4      | 0.10                        | -                                                              |
|                    | 4D        | 3.1 (1.4) | 3      |                             |                                                                |
|                    | 4A        | 3.5 (1.4) | 4      | 1.5                         |                                                                |
| Flood water        | 4B        | 4.3 (0.9) | 5      |                             |                                                                |
| overall condition  | 4C        | 4.0 (0.9) | 4      |                             | -                                                              |
|                    | 4D        | 3.7 (1.3) | 4      |                             |                                                                |
|                    | 4A        | 3.7 (1.5) | 4      |                             | 4A  vs  4B = 0.044                                             |
|                    | 4B        | 4.4 (0.8) | 5      |                             | 4A  vs  4C = 0.9                                               |
| Flood water level  | 4C        | 3.6 (1.2) | 4      | 0.02*                       | 4A  vs  4D = 0.6                                               |
| Tiood water level  | 4D        | 3.8 (1.3) | 4      | 0.02                        | 4B vs 4C = <b>0.008</b> *<br>4B vs 4D = 0.12<br>4C vs 4D = 0.4 |
|                    | 4A        | 3.5 (1.3) | 4      |                             |                                                                |
| T4                 | 4B        | 2.6 (1.6) | 2      | 1.0                         |                                                                |
| Trust on the crowd | 4C        | 2.8 (1.0) | 3      | 1.9                         | -                                                              |
|                    | 4D        | 2.9 (1.3) | 3      |                             |                                                                |

Participants expressed mixed attitudes toward the crowd. In some cases, some found comfort and direction in following others. In contrast, chaotic crowd behaviour or unclear group goals led some participants to intentionally diverge, expressing distrust or a preference for quicker or drier routes. Those who broke from the crowd often reported uncertainty about their decision, especially in the absence of social validation. Others described following the crowd reflexively, only later realizing the decision lacked conscious evaluation.

Familiarity with the environment was another major influential factor. Participants who had completed earlier trials felt more confident making independent decisions. Several acknowledged that without prior exposure, they likely would have followed the crowd, highlighting the role of repeated experience in reducing reliance on social cues. This was consistent with participants' expressions in VR1 to VR3.

Despite consistent environmental setups, nearly half of the participants reported perceiving changes in water depth or force. As in VR1-VR3, this indicates that subjective hazard perception, often guided by social behaviour, overrode the actual risk level. Some felt reassured after watching others cross safely, while others avoided water entirely based on general safety rules or instincts like "stay dry during floods."



Other influences included perceived distance, urgency, barriers, and sensory cues like sirens or flashing lights. While some participants chose longer dry paths for safety, others opted for quicker, riskier routes without fully assessing the conditions. A few disregarded environmental details entirely, acting on fixed rules or crowd movement alone.

Overall, VR4 findings reinforce the central role of flood severity in decision-making, with crowd influence remaining secondary and context-dependent, shaped by experience, trust, and clarity of the environment, echoing patterns from earlier VR studies.

#### 470 7 Discussion






This study examined how social cues, as the core focus of the study, shape evacuation decision-making in conjunction with environmental factors during flood events, using a series of four immersive VR experiments. Overall, study results consistently highlighted the dominant influence of social information, specifically crowd behaviour on both route choice and decision latency. These findings extend prior research demonstrating the power of social cues in shaping behaviour during emergencies (Helbing, Farkas et al. 2002, Petrucci 2022, Wang, Zhuang et al. 2024).

Across the first three experiments (VR1-VR3), findings demonstrate that human decision-making in flood evacuation is shaped more strongly by social dynamics than by the physical characteristics of the environment. While environmental hazards such as floodwater depth (moderate - around waist level) were rated as important, they did not consistently drive participants' behavioural responses. Instead, crowd-related cues, particularly behaviour, trust, and perceived intent, emerged as the primary determinants of both route selection and the timing of action. However, findings from the VR4 experiment introduce an important nuance: the moderating role of environmental severity. Specifically, while risky crowd behaviour in earlier scenarios typically reduced safe route selection, this influence weakened when floodwater levels were low (around ankle level). Risky behaviour only effectively discouraged participants from choosing the risky path when paired with high water levels (around shoulder/chest level), suggesting a boundary condition in which objective environmental risk can override or diminish the impact of social cues on evacuation behaviour.

Participants frequently relied on the behaviour of virtual crowds as heuristic indicators of safety, a pattern consistent with herding and social influence theories (Helbing, Farkas et al. 2002, Wang, Zhuang et al. 2024). When a crowd exhibited safe behaviour, individuals were significantly more likely to choose safer evacuation routes. Conversely, when crowds moved through risky flooded paths, participants had more tendency to follow them. This trend was especially apparent in conditions of uncertainty, such as when the destination was unclear (VR3) or the water level was low enough to seem doable (VR4). These patterns align with past work on herding behaviour in emergencies (Wang, Zhuang et al. 2024) and underscore the role of perceived group consensus in shaping individual risk perception.

One of the key insights from this research is the role of crowd size. While in VR1 participants frequently reported that crowd size affected their perceived risk, VR2 showed that the influence of a risky crowd was magnified when the crowd was large, reducing safe path selections and extending pre-movement times. While larger crowds sometimes conveyed safety and




legitimacy, they also introduced conflicting emotions such as anxiety or confusion and were described as dangerous and making the situation more competitive in participants' qualitative responses. These mixed perceptions echo previous findings that large groups can be interpreted as either protective or threatening depending on context (Haghani, Sarvi et al. 2019, Kinateder and Warren 2021). Notably, the presence of a large, safe-behaving crowd (2C) produced the most consistent influence toward safe decisions, suggesting that group reassurance can override individual uncertainty when social trust is high. However, crowd size is not significantly effective when the crowd exhibits risky behaviour.

Destination visibility played an important moderating role in how social cues were interpreted. In VR3, when the safe destination was clearly visible, participants were more confident in their choices and less reliant on crowd cues. In contrast, under ambiguous spatial conditions, such as when the destination was invisible and unknown, social influence became more pronounced. This aligns with spatial cognition research showing that environmental legibility reduces dependence on external cues (Gärling, Böök et al. 1986), and mirrors patterns observed in fire evacuation where visible exits promote faster and more accurate route selection (Fu, Liu et al. 2024).

Environmental factors, while consistently rated as influential, were experienced subjectively. Participants often perceived differences in water depth or flood severity despite uniform environmental setups across conditions. This finding reinforces evidence that hazard perception is socially modulated (Becker, Taylor et al. 2015, Bernardini, Camilli et al. 2017), with participants often drawing inferences from others' behaviour rather than direct appraisal of the physical environment. Even in VR4, where high water level deterred risky route selection more effectively than social cues alone, the influence of crowd behaviour remained context dependent. Risky behaviour was only rejected when environmental danger was both visible and unequivocally high, indicating a threshold where objective risk outweighed social signals.

Response latency (pre-movement time) was another important marker of internal conflict and uncertainty. In all studies except VR4, participants took significantly longer to act when exposed to risky social cues. These delays likely reflect cognitive conflict between intuitive social heuristics and rational risk (Adrian, Amos et al. 2025). VR4's lack of significant variation in pre-movement time may reflect an effect, where the physical extremity of floodwater made the danger so salient that social influence became secondary impactful factor on decision-making.

Qualitative data revealed additional layers of complexity. Some participants described following others instinctively, citing a need for social validation. Others consciously resisted social cues due to mistrust or prior knowledge of evacuation routes in between-subject design. Trust, frequently mentioned by participants, emerged as a crucial variable, varying across conditions and directly influencing decisions. Familiarity with the environment also moderated behaviour, with those in later trials more confident in ignoring misleading social cues, suggesting an adaptive carry over effect through repeated exposure.

Together, these findings advance current understanding of flood evacuation behaviour by integrating controlled experimental evidence with insights from real-world disaster psychology. The results highlight that evacuation models should not treat social and environmental factors as separate drivers, but as interdependent variables whose effects vary by context and perceived risk.






This research aligns with a growing body of work advocating for the incorporation of behavioural realism into flood risk modelling (Simonovic and Ahmad 2005, Alonso Vicario, Mazzoleni et al. 2020). The use of immersive VR in this study demonstrates its value in capturing the complex interplay between perception, cognition, and social context in flood scenarios, offering a safe, repeatable, and scalable method for generating empirical data to inform both theory and practice in natural hazards risk reduction.

Nonetheless, this study has several limitations. First, the sample size was small, and the participant demographic was limited, particularly in terms of age and education, which may affect the generalisability of the findings. Second, while VR offers a powerful tool for replicating real-world scenarios, questions remain regarding the reliability and ecological validity of behavioural data collected in such environments. Therefore, caution is warranted when interpreting the extent to which these findings apply to real-life flood evacuation behaviour.

Another limitation concerns the validation of the Likert scale questionnaire used to assess decision-making influences; further psychometric testing is needed to confirm its reliability and construct validity. Additionally, despite counterbalancing, the within-subject design may have introduced carryover effects, whereby participants' behaviour in later conditions was influenced by earlier exposures. Consequently, behavioural data from the second experimental condition onward should be interpreted with care, particularly in relation to social cue sensitivity and response consistency.

#### 8 Conclusions and Future Work

The aim of this study was to investigate the impact of social cue on human decision-making in flood in conjunction with other influential factors. The study results demonstrated human response during flood evacuation is heavily influenced by social cues, particularly crowd behaviour, often more so than by objective environmental risks such as floodwater depth or clarity on the safe destination. However, the impact of social cues was not static; it varied based on environmental context. While risky crowd behaviour generally encouraged participants to choose hazardous routes, this effect diminished when floodwater levels were visibly high indicating that objective environmental severity can override social influence. Crowd size and destination clarity further moderated decisions, highlighting the complex, context-sensitive nature of evacuation behaviour.

These findings underscore the importance of incorporating social dynamics into flood risk modelling, emergency planning, and public safety interventions. The use of VR demonstrates its effectiveness in capturing human behaviour under controlled yet realistic conditions, supporting the development of human-centred evacuation strategies. By examining how crowd behaviour shapes individual decision-making during flood evacuation, this research contributes to the interdisciplinary natural hazards risk reduction agenda and provides empirical evidence aligned with the scope of the Natural Hazards and Earth System Sciences journal.

Future research should aim to broaden participant diversity, including those from flood-prone or disaster-experienced communities, as their unique perspectives can offer critical insights into how prior experience and cultural context shape

https://doi.org/10.5194/egusphere-2025-5312 Preprint. Discussion started: 7 November 2025

© Author(s) 2025. CC BY 4.0 License.

EGUsphere

evacuation behaviours and decision-making under risk. Further, combining VR-derived behavioural data with modelling outputs like agent-based simulations and real-world analytics could enhance ecological validity and predictive power. Investigating long-term learning effects, emotional states, and collaborative behaviours within group evacuation contexts also presents valuable directions. Ultimately, improving our understanding of how people perceive and respond to flood threats can inform more adaptive, realistic, and effective flood evacuation protocols in an era of escalating climate hazards.

Ethical approval



This study was approved by the Ethics Committee of the Faculty of Engineering at the University of Nottingham.

**Author contributions statement** 

**Booloot Arshaghi:** Conceptualization, Methodology, Investigation, Writing - Original Draft, Visualization. **Glyn Lawson:** Supervision, Writing - Review & Editing. Conceptualization, Methodology, Funding acquisition. **Peer-Olaf Siebers:** Supervision, Reviewing. Conceptualization, Methodology, Funding acquisition. **Riccardo Briganti:** 

Supervision, Reviewing. Conceptualization, Methodology, Funding acquisition.

**Competing interests** 

The authors declare that they have no conflict of interest.

Acknowledgements

This work was supported by the Engineering and Physical Sciences Research Council (EPSRC) [grant number: EP/W524402/1].

Declaration of generative AI

During the preparation of this work the author(s) used ChatGPT in order to improve readability and language. After using this tool/service, the author(s) reviewed and edited the content as needed and take(s) full responsibility for the content of the publication.

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
