# Peer review of "Human Decision-Making in Crowds in a Virtual Flood Scenario"

_EGUsphere, 2025_

## Author Comment (AC1)

We would like to thank the reviewer for providing constructive comments to improve the manuscript. We provided a detailed answer to each comment in the following response.

1- Abstract. Consider changing 'a threshold where physical danger overrides social cues' to 'a threshold where more obvious physical danger overrides social cues.

We thank to the reviewer for their comments. We agree that specifying "more obvious" physical danger more accurately reflects the intended meaning. We revised the sentence in the abstract accordingly in the revised version of the manuscript. As per your suggestion, the updated wording is:

*"...a threshold where more obvious physical danger overrides social cues."*

2- Is it necessary to use the term 'herding', which is more suitable for animals and their instincts, has been criticised in the literature (see Haghani et al., 2019)? 'Social influence' (also used) is a more neutral alternative term.

Thank you for your insightful comment. We agree that the term herding may reflect instinctive and animal-like behaviour and note that it has been criticised in the literature. To avoid this unintended connotation, we replaced herding with alternatives including social influence and social cue (where appropriate), in the manuscript, which more accurately and naturally describe the deliberative, cognitively mediated processes suggested by our findings.

3- Observing the behaviour of the majority of people is a good guide for how one should behave (Gigerenzer, 2008), particularly when those people are judged to be self-relevant in some way (Spears, 2021).

We agree that these considerations are relevant to the scenarios of our VR studies. The participant will observe the crowd behaviour and include it in the individual decision process. We also agree that this influence is stronger when the observed people are self-relevant. Therefore, we have included these points and the citations in the discussion of the revised version of the manuscript.

4- The term 'natural disasters' is criticized in the disaster's literature, and the term 'hazards' is suggested instead (with disaster being the social effects of a hazard). See for example UNDRR https://www.undrr.org/our-impact/campaigns/no-natural-disasters.

We agree that the phrase '*natural disasters*' is not conceptually accurate and, for this reason, has been widely criticised in the literature. As highlighted in the UNDRR *No Natural Disasters* campaign, disasters are not "natural" events, but the outcome of a natural hazard interacting with social vulnerability, exposure, and insufficient protective measures.

In line with contemporary literature and the UNDRR position, we replaced the term natural disasters with natural hazards throughout the revised version of the manuscript to ensure that our terminology reflects the established conceptual distinction.

5- How was the interview data analysed?

Thank you for mentioning this. To address this comment and provide additional clarification on data collection and analysis, a more detailed description of the behavioural measures and data analysis has been added to the end of Procedure section as follow:

*"Participants completed a series of standard and custom questionnaires designed by authors at key stages of the experiment, including a demographic survey, the Simulator Sickness Questionnaire (Kennedy, Lane et al. 1993), the Igroup Presence Questionnaire (IPQ) (Schubert 2003), and a brief Likert-scale decision-making questionnaire designed by authors (appendix A), assessing the influence of environmental and social factors on route choice. Qualitative data were also collected through post-condition interviews (appendix B).*

*In addition to self-reported data, behavioural measures including path choice and pre-movement time were extracted for analysis after each experimental condition. Pre-movement time, also referred to as response time or pre-evacuation time, describes the interval between a stimulus and the initiation of action (Adrian, Amos et al. 2025). In this study, pre-movement time was defined as the interval between the onset of the simulated flood warning within the VR environment and the participant's initiation of evacuation movement. Specifically, it was measured as the duration spent observing and assessing the environment before initiating movement toward the designated safe destination and was extracted from VR screen recordings.*

*Quantitative data from questionnaires, behavioural measures, and pre-movement times were analysed using descriptive statistics to summarise central tendency and variability (mean/SD), followed by inferential statistical analyses to test differences across experimental conditions. Qualitative interview data were analysed using reflexive thematic analysis (Braun and Clarke, 2006), following an inductive approach. Interviews were transcribed verbatim and repeatedly reviewed to achieve familiarisation with the data. Meaningful segments related to participants' perceptions, decision-making processes, and experiences during the VR flood evacuation scenarios were systematically coded. Codes were then reviewed and grouped into candidate themes, which were iteratively refined through comparison across participants and experimental conditions. "*

Reference added:

BRAUN, V. & CLARKE, V. 2006. thematic analysis in psychology. Qualitative research in psychology, 3, 77-101.

6- Study 1. A large effect size is expected – but why? The fact that the authors suggest post hoc that the sample size was too small indicates that this assumption was unwarranted in this case.

Thank you for this comment. We agree that the assumption of a large effect size requires clarification. VR1 was designed as a feasibility study, with the primary aim of testing experiment task flow, VR usability etc. rather than conducting confirmatory hypothesis testing.

In this context, the assumed large effect size (f = 0.40) was used pragmatically to determine a minimum sample size, consistent with common research practice in pilot studies (Kunselman 2024). Our later observation that the sample size was insufficient to detect smaller effects does not contradict this assumption but rather demonstrates the intended limitation of VR1 and

informed the increased sample sizes used in VR2-VR4. We revised the manuscript (Participants, under Method section) to clarify this rationale, the revised paragraph is as follow:

*"VR1 was conducted as a feasibility study to assess procedural feasibility, VR usability, and the suitability of the experimental measures, and to guide the design of subsequent experiments. Accordingly, the study was not intended for confirmatory hypothesis testing or precise effect size estimation. A power analysis was conducted using GPower 3.1.9.7 (Faul et al., 2007) to estimate a minimum sample size, assuming a large effect (f = 0.40) for a one-way repeated-measures ANOVA with three conditions (α = 0.05, power = 0.80), resulting in a required sample of N = 12. "*

Reference:
Kunselman, A. R. (2024). "A brief overview of pilot studies and their sample size justification." Fertility and Sterility 121(6): 899-901.

**7-  Can the authors provide a link to footage/ moving visualisation?**

A link to the video footage of the VR experiment will be made available and stored on the University of Nottingham repository, following FAIR data sharing guidelines, or the Journal repositor permanently.

**8-  Was the questionnaire developed by the authors themselves or did they use established items? We should at least see some example items (and preferably there should be a link to the whole questionnaire, so the wording of items can be seen).**

The questionnaire was developed by the authors, and it is added to the Appendix of the revised version of the manuscript. It is also clarified in the revised manuscript, in the Procedure section, as below:

*"Participants completed a series of standard and custom questionnaires designed by authors at key stages of the experiment, including a demographic survey, the Simulator Sickness Questionnaire (Kennedy, Lane et al. 1993), the Igroup Presence Questionnaire (IPQ) (Schubert 2003), and a brief Likert-scale decision-making questionnaire designed by authors (appendix A), assessing the influence of environmental and social factors on route choice. Qualitative data were also collected through post-condition interviews (appendix B)."*

The Appendices have been modified as follows:

***"Appendix A-** After Experiment Questionnaires*

*The following table presents the questionnaires which participants completed after experiments. This questionnaire provided insight into the influence of decision factors on their decision making on choosing route to the safe destination in Likert scale.*

*Table A. Post Experiment Questionnaires*

| *- Please rate to what extent the following **factors** influenced your **decision-making** in **choosing your route** to the safe destination:    1= not at all to 5 = very high.`* | | | | | | *VR Experiment* |
|---|---|---|---|---|---|---|
| ***Decision Factors*** | *1* | *2* | *3* | *4* | *5* | |
| *Presence of the crowd* | | | | | | *1 – 2 – 3 – 4* |
| *Crowd choice of path* | | | | | | *1 – 2 – 3 – 4* |
| *Size of the crowd* | | | | | | *2* |
| *Flood water overall condition* | | | | | | *1 – 2 -3 – 4* |
| *Flood water level* | | | | | | *1 – 2- 3 -4* |
| *Trust on the crowd* | | | | | | *2- 3 -4* |

**Appendix B-** *Interview Questions**

*The following table (table B) shows the questions that participants responded to after the VR experiments*

*Table B. Interview Questions*

| *N* | *Questions* | *VR Experiments* |
|---|---|---|
| *1* | *Did you notice the people walking toward the safe destination? please explain. Do you believe that you were "consciously" following others/avoiding following them, to reach the destination/rescue team?* | *1 - 2 - 3 - 4* |
| *2* | *How did you assess that which route you need to go to reach the safe destination?* | *2- 3- 4* |
| *3* | *Did you notice the size of the crowd present in the scene? How was it? Did the size of the crowd affect your decision to the destination? please explain how.* | *2- 3- 4* |
| *4* | *Did you notice the level of water (water hight) before you chose your path to the destination? Did you have any concern to walk through the flood water?* | *1 – 2 – 3 -4* |
| *5* | *Were you aware of the risk of passing through the water first when you decided on your path to the safe destination? If yes, to what extend do you think it affect your action in this experiment?* | *1 - 2- 3- 4* |
| *6* | *Did you notice the distance from the destination when you were deciding which path you want to go through?* | *1 - 2- 3- 4* |
| *7* | *What other factors influenced your decision on choosing the path to the destination?* | *1 - 2- 3- 4* |
| *8* | *Do you think you could trust the crowd and the route they were taking to reach the safe destination?* | *2- 3- 4* |

*"*

9- Table 2 post hoc column seems to indicate that conditions were compared across experiments for VR3 and VR4, which is incorrect.

Thank you for this good point. This was a labelling mistake, which is corrected in the revised version of the manuscript as bellow:

| VR Study | Condition | Independent Variable | | | Participants Responses- Choice of Path | | | | Participants Pre-movement Time | | |
|---|---|---|---|---|---|---|---|---|---|---|---|
| | | Level1: Crowd Behaviour Value | Level 2 | Value | Response (Probability %) | | Cochran's Q Test | Post Hoc Pairwise Comparisons: McNemar Test | Mean (SD) | One-Way Repeated Measures ANOVA Test | Post Hoc Pairwise Comparisons |
| | | | | | Safe | Risky | P | P | | P | P |
| VR1 | 1A | Safe | N.A. | N.A. | 1.7 | 8.3 | 0.04* | 1A vs 1B = 0.125 | 7.5 4.5) | F(1.4,16.06)=11.8 p = 0.001* | 1A vs 1B =.025 |
| | 1B | Risky | | | 58.3 | 41.7 | | 1A vs 1C = 0.100 | 13.8(4.4) | | 1A vs 1C = 1.0 |
| | 1C | Control | | | 91.7 | 8.3 | | 1B vs 1C = 0.125 | 7.08(2.4) | | 1B vs 1C = 0.001* |
| VR2 | 2A | Safe | Crowd Size | Small | 95.8 | 4.2 | 0.006* | 2A vs 2B = 0.125 2A vs 2C = 1.00 2A vs 2D = 0.031 2B vs 2C = 0.125 2B vs 2D = 0.62 2C vs 2D = 0.03 | 8.6(5.2) | F (3,69) = 17.8 p = 0.001* | 2A vs 2B = 0.001* 2A vs 2C = 0.71 2A vs 2D = 0.001* 2B vs 2C = 0.001* 2B vs 2D = 0.14 2C vs 2D = 0.001* |
| | 2B | Risky | | | 79.2 | 20.8 | | | 13.5(5.1) | | |
| | 2C | Safe | | Large | 95.8 | 4.2 | | | 8.3(5.1) | | |
| | 2D | Risky | | | 70.8 | 29.2 | | | 15.4(7.3) | | |
| VR3 | 3A | Safe | Clarity on Location of the Safe Destination | Known | 100 | 0.0 | 0.001* | 3A vs 3B = 0.08 3A vs 3C = N. A 3A vs 3D = 0.001* 3B vs 3C = 0.008* 3B vs 3D = 0.125 3C vs 3D = 0.001* | 10.2(3.6) | F(3,63) = 18.4 p = 0.001* | 3A vs 3B = 0.05 3A vs 3C = 0.001* 3A vs 3D = 0.001* 3B vs 3C = 0.18 3B vs 3D = 0.001* 3C vs 3D = 0.001* |
| | 3B | Risky | | | 62.5 | 37.5 | | | 8.1(2.7) | | |
| | 3C | Safe | | Unknown | 100 | 0.00 | | | 11.9(2.9) | | |
| | 3D | Risky | | | 50 | 50 | | | 14.9(4.3) | | |
| VR4 | 4A | Safe | Flood Water Level | High | 100 | 0.0 | 0.001* | 4A vs 4B = 0.001* 4A vs 4C =0.001* 4A vs 4D =0.21 4B vs 4C = N.A 4B vs 4D = 0.008* 4C vs 4D = 0.008* | 8.3(4.5) | F(3,66) =14.8 p = 0.28 | - |
| | 4B | Risky | | | 79.1 | 20.8 | | | 9.4(4.6) | | |
| | 4C | Safe | | Low | 83.3 | 16.6 | | | 7.9(3.9) | | |
| | 4D | Risky | | | 33.3 | 66.6 | | | 9.5(4.8) | | |

Note that the original table has been modified, in order to make it more concise and readable.

10- Questionnaire tables should include notes reminding us what the A, B, C, D conditions are.

Thanks for your suggestion. The descriptions of labels are now provided in the caption of all Decision Factors results table as presented for Table C below. Please note that due to suggestion made by the Reviewer2, these tables are moved to appendix to increase the readability of the paper.

*"**Table C:** VR1 Decision Factors Questionnaire Results (1A= Crowd Behaviour: Safe; 1B = Crowd Behaviour: Risky; 1C = No Crowd). "*

11- Page 18. It is unclear what is meant by 'chaotic crowd behaviour' in the analysis of study 4, as there is no indication earlier that 'chaotic behaviour' would be varied in the

This term was reported by participants when describing crowd behaviour and was therefore intended to be included as a quotation. However, due to the revision to the Discussion section, this part, including the term, has been removed.

12- The discussion makes a strong claim that social cues are more important than environmental cues, even for deep floodwater. However, the analysis of VR4 could make it much more clear whether there was a significant main effect of flood water level (rather than just the interaction/ tests across the four conditions).

We agree that the original wording of the Discussion could be interpreted as overstating the dominance of social cues, without sufficiently distinguishing the main effect of floodwater level in VR4 study. We revised the structure of discussion and divided it into main three subsections (suggested by the Reviewer2) for more clarity and readability. To address this comment, we also revised the relevant parts in this section to clarify that floodwater level exerted a meaningful main effect on route choice in VR4, with high water levels significantly discouraging risky route

selection. The revised text below (in bold) emphasises that social cues strongly influence behaviour under conditions of uncertainty or moderate risk, but that their influence is constrained when environmental danger becomes visibly severe. These revisions better reflect the VR4 findings and clarify the interaction between social and environmental factors The following demonstrates the final changes in the discussion in Bold for more clarification:

*"4 Discussion*

*4.1 Key Findings*

*This study examined how social cues shape evacuation decision-making during flood events using a series of four immersive VR experiments. Overall, the results consistently highlighted the* ***strong influence*** *of social information, particularly crowd behaviour, on both route choice and decision latency, extending prior research on social influence in emergencies (Helbing, Farkas et al. 2002, Petrucci 2022, Wang, Zhuang et al. 2024).*

*Across the first three experiments (VR1–VR3), decision-making was* ***shaped more dominantly by social dynamics than by physical environmental characteristics. Although the physical hazard indicators, such as floodwater depth (moderate, around waist level), were rated as important, they did not consistently produce independent behavioural effects****. Instead, crowd-related cues, particularly behaviour, trust, and perceived intent, emerged as the primary determinants of route selection and pre-movement time.*

*Findings from the VR4 experiment introduce an important nuance by highlighting the role of environmental severity. Specifically,* ***floodwater level exerted a meaningful effect on route choice****, with high water levels (around shoulder/chest level) significantly discouraging risky route selection. Importantly, this effect* ***interacted with social cues****: while risky crowd behaviour in earlier scenarios typically reduced safe route selection, this influence weakened when floodwater levels were low (around ankle level) and became substantially constrained when floodwater was visibly deep. This suggests a boundary condition in which objective environmental risk can override or diminish the impact of social cues on evacuation behaviour.*

*Participants frequently relied on the behaviour of virtual crowds as heuristic indicators of safety, consistent with following others and social influence theories (Helbing, Farkas et al. 2002, Wang, Zhuang et al. 2024). Safe crowd behaviour increased selection of safer routes, while risky crowd behaviour encouraged following flooded paths, particularly under uncertainty, such as unclear destinations (VR3) or seemingly manageable water levels (VR4). These patterns align with previous work on following others in emergencies and highlight the role of perceived group consensus in shaping individual risk perception (Wang, Zhuang et al. 2024).*

*Crowd size also played a moderating role. While participants in VR1 frequently reported that crowd size affected perceived risk, VR2 showed that large crowds amplified social influence primarily when exhibiting safe behaviour. Large risky crowds increased uncertainty, reduced safe route selection, and prolonged pre-movement time, while also eliciting mixed emotional responses. These findings echo evidence that large groups can be perceived as either protective or threatening depending on context (Haghani, Sarvi et al. 2019, Kinateder and Warren 2021). Notably, large safe-behaving crowds produced the most consistent shift toward safe decisions, whereas crowd size alone was less effective when behaviour was risky. This pattern suggests that participants did not follow the majority automatically but instead evaluated the observed behaviour of others as informative cues for action, particularly when the crowd was perceived as relevant or credible.*

*Destination visibility further moderated reliance on social cues. In VR3, visible safe destinations increased confidence and reduced dependence on crowd behaviour, whereas ambiguous spatial conditions intensified social influence. This aligns with spatial cognition research showing that environmental legibility reduces reliance on external cues (Gärling, Böök et al. 1986) and mirrors findings from fire evacuation studies (Fu, Liu et al. 2024).*

*Environmental factors were consistently rated as influential but were experienced subjectively. Participants often inferred flood severity from others' behaviour rather than direct appraisal, reinforcing evidence that hazard perception is socially modulated (Becker, Taylor et al. 2015, Bernardini, Camilli et al. 2017). Even in VR4, where high water levels discouraged risky choices more effectively, social influence remained context dependent.*

*Pre-movement time reflected internal conflict and uncertainty. In all studies except VR4, participants took significantly longer to act when exposed to risky social cues. VR4's lack of significant variation in pre-movement time may reflect a __boundary effect__, whereby the physical extremity of floodwater made the danger sufficiently salient that social influence played a secondary role in shaping response timing."*

13- Adrian, J., M. Amos, C. Appert-Rolland, M. Baratchi, N. Bode, M. Boltes, T. Chatagnon, M. Chraibi, A. Corbetta and A. Cuesta (2025). "Glossary for Research on Human Crowd Dynamics. This needs to be properly cited as the second edition.

The reference has been corrected in the revised version of the manuscript as shown below:

ADRIAN, J., AMOS, M., APPERT-ROLLAND, C., BARATCHI, M., BODE, N., BOLTES, M., CHATAGNON, T., CHRAIBI, M., CORBETTA, A. & CUESTA, A. 2025. Glossary for Research on Human Crowd Dynamics. *Collective Dynamics,* Second edition 1-32.